# The Clinical and Prognostic Characteristics of Primary Salivary Gland-Type Carcinoma in the Lung: A Population-Based Study

**DOI:** 10.3390/cancers14194668

**Published:** 2022-09-25

**Authors:** Lei-Lei Wu, Jia-Yi Qian, Chong-Wu Li, Yu Zhang, Wei-Kang Lin, Kun Li, Zhi-Xin Li, Dong Xie

**Affiliations:** 1Department of Thoracic Surgery, Shanghai Pulmonary Hospital, School of Medicine, Tongji University, Shanghai 200433, China; 2School of Medicine, Tongji University, Shanghai 200092, China

**Keywords:** salivary gland-type carcinoma, proportion, survival, surgery, lung cancer

## Abstract

**Simple Summary:**

According to reports from more than a decade ago, the proportion of primary salivary gland-type carcinoma (SGC) among all lung cancers is only 0.1–1.0% and the 5-year overall survival rate is more than 60%. However, previous reported studies mostly had small sample sizes due to the low proportion of primary SGC in lung cancer. The characteristics of SGC proportion and prognosis have not yet been elucidated. The aim of this study was to elucidate the clinical and prognostic characteristics of primary SGC. This study found that lung SGC has the best prognosis among adenocarcinoma, squamous cell carcinoma, and SGC. In addition, lobectomy can further improve the prognosis of SGCs.

**Abstract:**

This study aimed to explore the clinical and prognostic characteristics of primary salivary gland-type carcinoma (SGC). The entire cohort from the Surveillance, Epidemiology, and End Results database was used to calculate the SGC proportion. In total, 253,096 eligible patients, including 165,715 adenocarcinomas (ADCs), 87,062 squamous cell carcinomas (SCCs), and 319 SGCs, were selected to perform survival analyses. The data of 42 SGC patients from our hospital showed postoperative survival. Overall survival (OS) curves for different histological and surgical types were presented. The proportion of primary SGCs was 0.8 per 1000 patients. Patients with age ≤ 64 years old had a much higher proportion of SGC than those patients with age >64 years old. After adjusting for other confounders, among ADCs, SCCs, and SGC, SGCs had the best prognosis (HR 0.361, *p* < 0.001). Moreover, the 5-year OS rates of SGC patients were 55% and 7% in the group with surgery or without surgery, respectively (*p* < 0.001). The data of 42 patients from our hospital also showed a good survival of SGCs. Lobectomy improved the survival of SGCs significantly (adjusted HR 0.439, *p* = 0.016). In conclusion, pulmonary SGCs had the best prognosis among ADCs, SCCs, and SGCs. In addition, lobectomy could further improve the prognostic outcomes of SGCs.

## 1. Introduction

Lung cancer is still the leading cause of cancer mortality worldwide, although the morbidity rate has decreased [1,2]. In all lung malignancies, non-small-cell lung cancer (NSCLC) and small-cell lung cancer (SCLC) account for about 85% and 15%, respectively [3]. Salivary gland-type carcinomas (SGCs) originate from the salivary glands located in the upper aerodigestive tract, such as parotid, submandibular, and sublingual [4,5,6]. SGCs mainly include mucoepidermoid carcinoma (MEC), adenoid cystic carcinoma (ACC), and epithelial-myoepithelial carcinoma (EMC) [7].

According to reports from more than a decade ago, however, the proportion of the primary SGCs is only 0.1–1.0% in all lung cancers [8,9]. Previous studies suggested that lung cancer patients for SGC had satisfactory outcomes, of which the 5-year overall survival (OS) rate was over 60% [7,10]. Due to the low proportion of primary SGCs in lung cancer, previously reported studies mostly had small sample sizes [7,10,11]. Thus, the information about the proportion of primary SGCs in all lung malignancies needs to be updated. Some research has analyzed postoperative survival [7,10]; however, the role of surgery in the prognosis of SGCs needs to be further explored. In addition, it is unclear what the better prognostic outcomes in SGCs and other NSCLCs are. Therefore, it is important to study the surgical significance of SGCs and compare differences in survival between SGC and other NSCLCs, such as adenocarcinoma (ADC) and squamous cell carcinoma (SCC). The aim of this study was to investigate the clinical and prognostic characteristics of patients with SGCs.

## 2. Materials and Methods

### 2.1. Patients

The Ethics Committee of Shanghai Pulmonary Hospital approved this study (IRB number K22-209). Cases were diagnosed as lung malignancy in the Surveillance, Epidemiology, and End Results (SEER) database, which contains clinicopathological and survival data of cancer patients from 18 registries [12]. Patients diagnosed as primary SGC in Shanghai Pulmonary Hospital were also included in this study. All patient records were anonymized before analysis. The information about eligible patients was used to perform proportion and prognostic analyses. The selection criteria of patients are presented in Figure 1. The data for a total of 593,662 patients between 2004 and 2018 was used in the proportion calculation, and the data of 253,096 cases between 2004 and 2015 was used for survival analyses. In addition, we collected 42 SGC patients with surgery between 2011 and 2018 in Shanghai Pulmonary Hospital. These 42 patients did not have distant metastasis of the lymph nodes or other organs. Information collected from the SEER database included race/ethnicity, sex, age at diagnosis, tumor location, marital status, treatment approach (including surgical treatment, radiotherapy, and chemotherapy), tumor size, tumor differentiation, histological subtype, tumor node metastasis (TNM) stage, survival time, and definitive survival status. According to the staging-related information and the guidelines of the 8th TNM staging system [13], the 6th TNM stage was re-translated to the 8th edition. Cases from 2016 to 2018 were excluded from further analyses due to a lack of details required for retranslation from the sixth into the eighth edition staging.

### 2.2. Follow-Up

The follow-up duration of the cases for the survival analyses ranged from 0.0 to 179.0 months, with a median of 13.0 months. The follow-up information of the SEER database was updated in November 2020. The patients included in the survival analyses had definitive survival status, death or alive. OS, which was the duration from the date of diagnosis to death, was regarded as our observational endpoint.

### 2.3. Statistical Analysis

The main statistical analyses were performed using SPSS statistics 25.0 software (IBM SPSS Inc., Chicago, IL, USA), R 4.1.2 software (R Fundamental for Data Science, Vienna, Austria) (https://www.r-project.org/, accessed on 1 December 2021), and GraphPad Prism 9 (https://www.graphpad.com/scientific-software/prism/, accessed on 7 December 2021). Categorical variables were compared using Pearson’s Chi-square test. The analysis of linear regression was used to identify the relation between the proportion of SGC and the year of diagnosis. Risk ratios (RRs), hazard ratios (HRs), and 95% confidence intervals (CIs) were calculated using logistic regression analysis and Cox regression analysis, respectively (the regression method was Enter selection). Kaplan–Meier survival analysis and the Log-rank test were used to draw and compare the survival curves. The multivariable Cox proportional-hazards model was used to calculate the average value of each covariate and estimate the adjusted survival curves of different surgical types [12]. Statistical tests were considered statistically significant with a two-sided *p* value < 0.05.

## 3. Results

### 3.1. Patient Characteristics

In the cohort for the proportion analyses, the majority of the patients were over 64 years old (N = 402,390, 67.8%). The majority of the patients were male (N = 312,706, 52.7%), and 480,358 were Caucasian patients (80.9%). In total, 462 patients were diagnosed with SGCs, including 270 MECs, 181 ACCs, and 11 EMCs.

After case selection, 253,096 eligible patients, including 319 SGCs, 165,715 ADCs, and 87,062 SCCs, were included in the survival analyses. There were 122 ACCs, 193 MEC, and 4 EMCs in all 319 SGCs. The majority of cases were stage IV (N = 119,382, 47.2%), and the rate of stage IIA was low at 2.2% (N = 5676). The tumor differentiation of 104,001 patients was undifferentiated grade (41.1%), and 71,327 patients had a poor grade (28.2%). In total, 69.0% of the patients were confirmed not to undergo an operation (N = 174,529). In total, 108,109 patients (42.7%) received chemotherapy, and 98,998 patients (39.1%) underwent radiotherapy. Of note, 50,359 patients did not undergo any treatment. Other detailed information on the patient characteristics from the SEER database and our hospital is shown in Table 1 and Table 2, respectively.

### 3.2. The Analyses of Proportion

In the entire cohort of 593,662 patients, 462 cases were diagnosed as primary SGCs. The proportion was 0.8 (95% CI 0.3–1.3) per 1000 patients, which was essentially unchanged from 2004 to 2018 (Figure 2, 0.7 [95% CI 0.4–1.0] per 1000 persons in 2004; 1.1 [95% CI 0.8–1.4] per 1000 persons in 2018). The results of the linear regression also revealed that the proportion of SGCs was not changed by the year of diagnosis (r = 0.432, *p* = 0.108). The RR (diagnosis in 2018 vs. diagnosis in 2004) was 1.492 (95% CI 0.916–2.430, *p* = 0.108), adjusted for sex, age, and race. The results of the multivariable logistic regression revealed that patients with age ≤ 64 years had a much higher proportion of SGC than patients with age > 64 years (adjusted RR = 0.341, 95% CI 0.283–0.410, *p* < 0.001, Figure 2), and sex and race/ethnicity did not have an impact on the proportion of primary SGCs (all *p* > 0.05).

### 3.3. Survival Analysis of Different Histological Types

The median survival time of 253,092 patients was 13.0 months (ranging from 0.0 to 179.0 months). Moreover, the 1-, 3-, and 5-year OS rates of this cohort were 38%, 25%, and 9%, respectively. The unadjusted 5-year OS rate was best in patients with SGCs (41%) and worst in patients with SCCs (7%). The median survival time was 121.0 months (95% CI 83.8–158.2 months) in patients with SGCs, 14.0 months in ADC patients, and 12.0 months in SCC patients, which indicated that SGC patients had satisfactory outcomes (Figure 3A). Multivariable Cox regression analysis was used to identify the prognostic role of SGCs in the different NSCLCs (Table 3). After adjusting for other confounders, among patients with ADCs, SCCs, and SGCs, SGCs had the best prognosis (adjusted HR 0.513, 95% CI 0.437–0.603, *p* < 0.001) and SCCs had the worst survival (adjusted HR 1.119,95% CI 1.108–1.130, *p* < 0.001). Furthermore, the entire primary SGC cohort had much better survival than patients with stage IB ADCs or stage I SCCs (Figure 3B).

### 3.4. Surgical Significance in Primary SGCs

To further investigate the surgical significance for the prognosis of primary SGCs, we first combined patients with lobectomy, sub-lobectomy, and pneumonectomy into one group. Two patients were excluded from the analyses because they lacked detailed information about whether they had undergone surgery. The survival curves showed that SGC patients who underwent surgery had much better survival than patients who did not undergo surgery (Figure 3C, *p* < 0.001). The median survival time in the SGC patients without surgery was 12.0 months (95% CI 5.1–18.9 months); however, the median survival time was not reached in cases with surgery. There were 65 death events in the surgery group, which accounted for 30.1%. However, 81.2% of the patients in the group without surgery reached the end of life. The 5-year OS rate of the patients was 55% (Figure 4A) and 7% in the group with surgery and without surgery, respectively (unadjusted HR = 0.137, 95% CI 0.096–0.195, *p* < 0.001). In addition, there were 42 SGC patients and only 5 patients with recurrent or metastatic diseases between 2012 and 2017 in our hospital (Table 2). The duration of follow-up ranged from 39 to 129 months (median, 70.5 months). The data of our hospital recorded that all pulmonary SGC patients who underwent operations were alive (follow-up update on 23 March 2022). The 5-year disease-free survival rate was 88.0% in the Shanghai Pulmonary Hospital cohort (Figure 4B).

A prognostic analysis for postoperative patients was further performed. In the SEER database, of 216 SGC patients who underwent surgery, 5 patients were excluded from prognostic analysis because of their distant metastasis of lymph nodes or other organs. After adjusting for other confounders, the multivariable analysis identified poor to undifferentiated grade, age > 64 years, N1 classification, and tumor size of 3.1–5.0 cm as independent risk factors for postoperative prognosis (Table 4). Lobectomy, as an independent protective factor, improved the survival outcomes of SGCs significantly (adjusted HR = 0.439, 95% CI 0.225–0.856). The 5-year OS rates were 45% vs. 65% in the cases of sub-lobectomy and lobectomy, respectively (Figure 4C, overall *p* = 0.005).

## 4. Discussion

In the present study, the data of 593,662 patients was used to perform proportion analyses. We found that the proportion of primary SGCs in the lung was 0.8 per 1000 patients based on the SEER database between 2004 and 2018, which is consistent with previous reports [4,8]. The results also revealed that the proportion of SGCs did not change over time. Moreover, the incidence rate differed by age group. Patients with age ≤ 64 years had a higher proportion of primary SGCs than cases with age > 64 years. Moreover, sex and race/ethnicity did not have an impact on the incidence rate of primary SGCs. To further explore the prognosis of SGCs, we compared the survival among ADCs, SCCs, and SGCs. In the NSCLCs, previous reports about the comparison of prognoses among ADCs, SCCs, and SGCs are lacking. The multivariable Cox regression uncovered that SGC patients had the best survival, and SCC patients had the worst survival among the three classifications of patients. In the SGC group with operation, we investigated the surgical significance for prognosis. We confirmed that older patients had a poorer prognosis than younger patients, and lobectomy could improve the survival outcomes. In addition, the prognosis for the group of ACCs and EMCs was poorer than MECs. After adjusting for other confounders, grade, age, surgical types, N classification, and tumor size were identified as independent factors affecting survival. Given the surgery patients, especially lobectomy, have satisfactory survival outcomes, we propose that primary SGC patients without metastatic disease are recommended for surgery. Furthermore, lobectomy is more optimal than others.

The impact of the grading of tumors on the prognosis of primary SGCs remains unclear. A previous report found that high-grade SGCs were likely to present lymph-node metastasis [14]. In addition, some studies showed that high-grade primary SGCs had much poorer survival than low- to intermediate-grade cases [7,15]. Our results in the present study, however, revealed that poor- to undifferentiated-grade SGCs had a poorer prognosis than those with well to moderate grades. The findings from Hsieh CC et al. showed that the grade of tumor did not affect the survival of pulmonary MECs [16]. The results in the above studies are not consistent. On the one hand, there were other studies about pulmonary SGCs; however, survival analyses were lacking [17,18]. Moreover, the sample size of primary SGCs was small, and the information about tumor grading was not available in some cases [17]. On the other hand, many high-grade neoplasms can show the presence of low-grade areas even though they do not form the bulk of the tumor [19]. Thus, the prognostic significance of tumor grading in pulmonary SGCs needs to be further investigated.

Surgery, especially lobectomy, should be recommended for eligible patients of primary SGCs. A study from Kim BG et al. showed that surgical resection could improve survival of primary SGCs compared with bronchoscopic intervention based on the analyses of information from 181 patients [20]. Another study by Zhu F et al. also revealed that patients who underwent operation had satisfactory survival outcomes [7]. In the present study, we found similar results to the abovementioned two studies. Regrettably, the above two studies did not compare the effect of different surgical approaches on prognosis with pulmonary SGCs. We found that patients who underwent lobectomy had improved survival compared to sub-lobectomy. In addition, in the univariable analysis, the N1 and N2 classifications were regarded as factors affecting survival. After multivariable analysis, however, only one of the above two factors, N1 classification, could independently affect prognosis. N2 classification is a prognostic risk indicator but did not present statistical significance in the multivariable analysis. The reason for this phenomenon might be that the number of patients with N2 was too small. Therefore, we think that the resection of lymph nodes during operation for pulmonary SGCs still needs to be focused on.

In addition, the histological type of ACC served as a negative factor influencing survival compared with MECs. The findings of the research by Wang Y et al. showed that OS in ACC patients was lower than that in MEC patients [21]. This study included 108 MEC patients and 183 ACC patients. The eligible patients of their research mainly underwent tracheobronchial resection. Patients who underwent lobectomy operations were the majority in another study from Zhu Y et al. [7]. They agreed on the results of the above research that MECs had better survival than ACCs; however, they did not perform multivariable analysis to adjust for confounders, including age, sex, TNM stage, and tumor differentiation. Thus, the results of the research from Zhu Y et al. might be challenged, as they were affected by the failure to consider other confounding factors. The present study included 63 ACCs, 147 MECs, and 1 EMC in the SEER cohort with surgery. In the univariable analysis, we found similar results to the above two studies. As the sample size was small, the comparison of survival in the different histological subtypes did not have statistical significance after multivariable Cox regression. Based on this, we still believe that the prognosis of MEC may show a better trend than that of ACC. In addition, prognostic research on EMC, as the least common tumor in SGCs, needs to be put on the agenda.

There was an interesting phenomenon in the results in that marital status had a prognostic effect on NSCLS patients. Married patients had a better survival than non-married patients, which is in accordance with some studies involving other malignant tumors [22,23]. Several possible reasons may explain the relationship between marital status and survival outcomes. First, given that married patients are more likely to receive support from family members, they may be diagnosed much earlier and be inclined to receive treatment, leading to a satisfactory survival rate. Second, mental support is urgently needed for cancer patients in addition to physical care. Married patients often receive their spouses’ help throughout diagnosis and treatment. However, unpleasant and upsetting relationships generate depression, which may have acted as a predictor of disease progression and motility of malignant tumors in a meta-analysis [24]. Without support from spouses, unmarried patients are prone to suffering from greater emotional pressure and worse socioeconomic situations, which may be associated with poor prognosis among patients with NSCLC.

Notably, patients with stage I SCCs had a poor prognosis. In previous studies, some studies compared the prognosis of ADCs and SCCs and found that the prognosis of ADCs is better than that of SCCs [25,26]. In one report, non-cancer-related deaths in patients with SCCs mainly included pneumonia, chronic obstructive pulmonary disease, and sepsis [25]. Moreover, this study found that SCC patients had a higher non-cancer-related mortality rate than ADC patients [25]. The observational end point of our study was OS, which might result in a lower OS rate in SCC patients because of its high non-cancer mortality rate. In addition, considering that this study was a retrospective study, the data on staging in the SEER database might not be detailed and accurate enough, which would cause staging differences to a certain extent, so that relatively advanced patients enter the cohort with stage I. However, ADC patients have a better prognosis than SCCs overall. Therefore, even if it is early stage SCC, we should pay attention to its prognosis and give necessary postoperative adjuvant therapy and follow-up management.

This study has several limitations. First, some important information was not detailed, such as the completeness of resection, chemotherapy sequence, and immunohistochemistry, as we could not obtain it from the SEER database. Second, although the data we used was obtained from a large population-based cohort, the sample size of pulmonary SGCs was still small. For example, we categorized patients with wedge resection or segmental resection into sub-lobe resection because of the small sample of these two kinds of patients. Third, we excluded patients with age < 20 years to ensure that all patients were not juveniles. However, given that age in the SEER database is not a continuous value, it is grouped at five-year intervals (such as 15–19 and 20–24), so patients aged 18–19 could only be excluded from this study. This action might affect the results of statistical analyses. Finally, given that this study was a retrospective study, it was impossible to avoid selection bias. Therefore, more studies are necessary to further validate our findings.

## 5. Conclusions

Primary SGCs in all lung cancers were rare, of which the incidence rate was 0.8 per 1000 patients between 2004 and 2018. Moreover, patients with age ≤ 64 years had a much higher proportion of SGC than patients with age > 64 years. Fortunately, pulmonary SGCs had satisfactory survival, and they had the best prognosis among ADCs, SCCs, and SGCs. In addition, surgery, particularly lobectomy, could further improve the prognostic outcomes of SGCs. However, more research is required to confirm these findings.

## Figures and Tables

**Figure 1 cancers-14-04668-f001:**
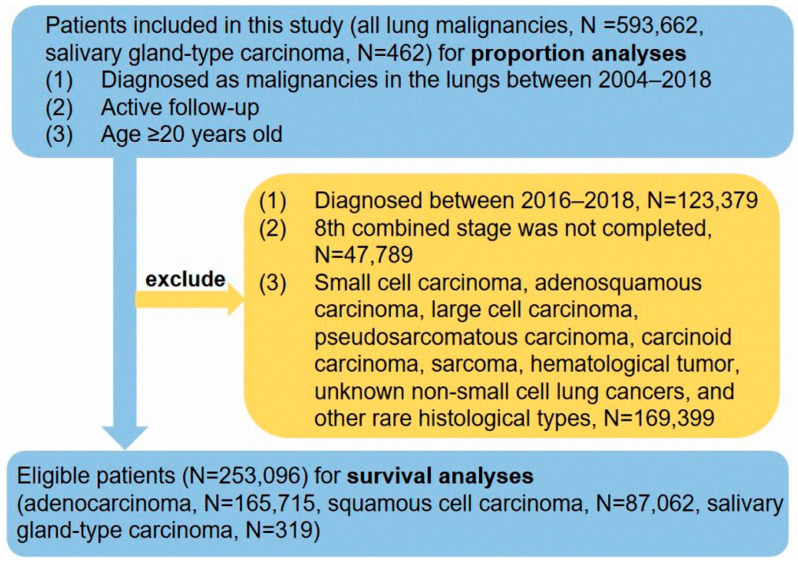
The flow chart of this study.

**Figure 2 cancers-14-04668-f002:**
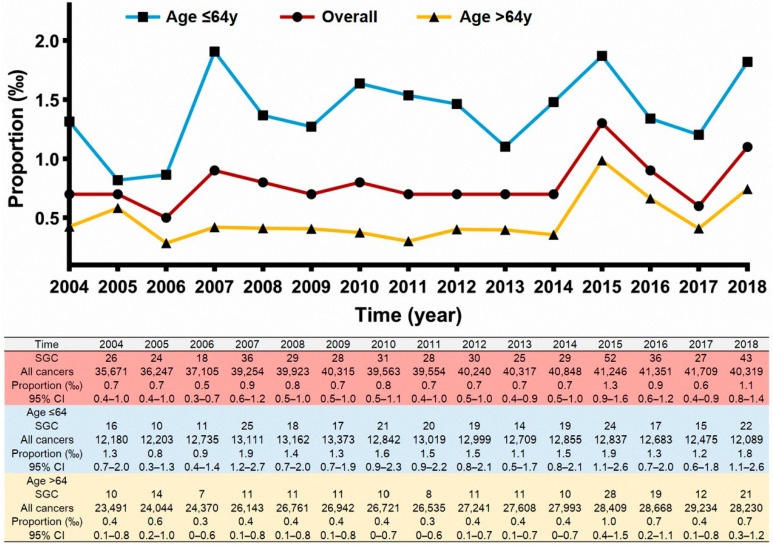
The crude incidence rate of salivary gland-type carcinoma over time in the 593,662 lung cancer patients.

**Figure 3 cancers-14-04668-f003:**
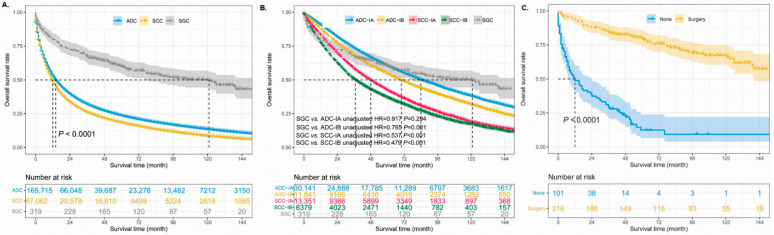
The unadjusted survival curves of different histological types (**A**,**B**). The unadjusted survival curves for SGC patients according to whether they received surgery (**C**). ADC: adenocarcinoma, SCC: squamous cell carcinoma, SGC: salivary gland-type carcinoma.

**Figure 4 cancers-14-04668-f004:**
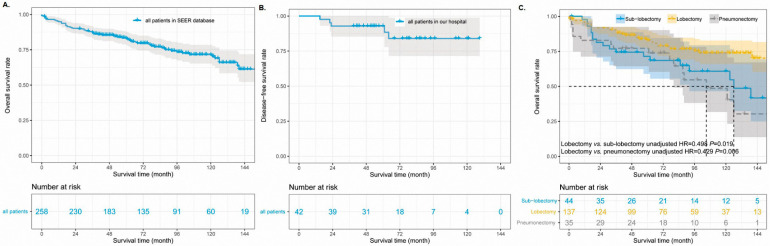
The unadjusted survival curves of for patients who underwent surgery in the SEER database (**A**) and Shanghai Pulmonary Hospital (**B**). The adjusted survival curves of different surgical types (**C**). SEER: Surveillance, Epidemiology, and End Results.

**Table 1 cancers-14-04668-t001:** Clinical and pathological characteristics of lung cancer patients from the Surveillance Epidemiology and End Results database between 2004 and 2015.

Variables	ADC	SCC	SGC	*p* Value
Total	165,715	87,062	319	
Sex (%)				<0.001
Male	80,619 (48.6)	55,030 (63.2)	158 (49.5)	
Female	85,096 (51.4)	32,032 (36.8)	161 (50.5)	
Age (%)				<0.001
≤64	59,635 (36.0)	24,745 (28.4)	198 (62.1)	
>64	106,080 (64.0)	62,317 (71.6)	95 (37.9)	
Race (%)				<0.001 *
Caucasian	130,775 (78.9)	71,277 (81.9)	245 (76.8)	
Other	34,762 (21.0)	15,725 (18.1)	71 (22.3)	
Unknown	198 (0.1)	60 (0.1)	3 (0.9)	
Marital status (%)				<0.001
Non-marital	71,452 (43.1)	39,464 (45.3)	116 (36.4)	
Marital	87,207 (52.6)	44,072 (50.6)	194 (60.8)	
Unknown	7056 (4.3)	3526 (4.1)	9 (2.8)	
Grade (%)				<0.001
Well	15,631 (9.4)	1761 (2.0)	66 (20.7)	
Moderate	35,835 (21.6)	24,387 (28.0)	88 (27.6)	
Poor	40,757 (24.6)	30,516 (35.1)	54 (16.9)	
Undifferentiated	73,492 (44.4)	30,398 (34.9)	111 (34.8)	
TNM stage (%)				<0.001
IA	30,141 (18.2)	13,351 (15.3)	121 (37.9)	
IB	11,841 (7.1)	6379 (7.3)	47 (14.7)	
IIA	2853 (1.7)	2816 (3.2)	7 (2.2)	
IIB	8521 (5.1)	6790 (7.9)	32 (10.1)	
IIIA	14,771 (8.9)	13,153 (15.1)	31 (9.7)	
IIIB	10,397 (6.4)	12,445 (14.3)	18 (5.6)	
IV	87,191 (52.6)	32,128 (36.9)	63 (19.7)	
Tumor location (%)				<0.001
Upper lobe	87,138 (52.6)	47,121 (54.1)	93 (29.2)	
Middle lobe	7574 (4.6)	3199 (3.7)	31 (9.7)	
Lower lobe	44,625 (26.9)	25,263 (29.1)	104 (32.6)	
Others	3676 (2.2)	4655 (5.3)	56 (17.5)	
Unknown	22,702 (13.7)	6824 (7.8)	35 (11.0)	
Surgical approach (%)				<0.001 *
None	112,351 (67.8)	62,077 (71.3)	101 (31.7)	
Sub-lobectomy	12,449 (7.5)	4865 (5.6)	44 (13.8)	
Lobectomy	38,686 (23.3)	17,437 (20.0)	137 (42.9)	
Pneumonectomy	1518 (0.9)	2209 (2.5)	35 (11.0)	
Unknown surgical approach	254 (0.2)	120 (0.1)	1 (0.3)	
Unknown	457 (0.3)	354 (0.5)	1 (0.3)	
Radiotherapy (%)				<0.001 *
None	105,559 (63.7)	46,485 (53.4)	234 (73.4)	
Yes	59,042 (35.6)	39,874 (45.8)	82 (25.7)	
Unknown	1114 (0.7)	703 (0.8)	3 (0.9)	
Chemotherapy (%)				<0.001
None	93,791 (56.6)	50,920 (58.5)	276 (86.5)	
Yes	71,924 (43.4)	36,142 (41.5)	43 (13.5)	

ADC: adenocarcinoma, SCC: squamous cell carcinoma, SGC: salivary gland-type carcinomas. Categorical variables were compared using Pearson’s Chi-square test. * Fisher’s exact test.

**Table 2 cancers-14-04668-t002:** Detailed information about pulmonary salivary gland-type carcinoma in Shanghai Pulmonary Hospital.

Case	Sex	Histological Subtype	Grade	Age	Surgical Approach	Complete Resection	Adjuvant Therapy	TNM Stage	Postoperative Hospital Stay (Day)	Type of Resection	Smoking History	DFS (Month)
1	Male	MCE	Low	34	Open	Yes	Radiotherapy	IA	6	Sub-lobectomy	Never	42.00
2	Male	MCE	Low	44	VATS	Yes	None	IA	5	Lobectomy	Never	41.00
3	Female	ACC	-	65	VATS	Yes	None	IA	5	Lobectomy	Never	74.00
4	Female	MCE	Low	59	Open	Yes	None	IA	6	Lobectomy	Never	60.00
5	Male	MCE	Low	60	VATS	Yes	None	IA	6	Lobectomy	Never	54.00
6	Female	MCE	-	30	VATS	Yes	None	IA	6	Lobectomy	Never	112.00
7	Male	MCE	Low	44	VATS	Yes	None	IB	3	Sub-lobectomy	Never	55.00
8	Female	MCE	-	69	VATS	Yes	None	IB	4	Lobectomy	Never	39.00
9	Male	MCE	-	45	Open	Yes	None	IB	12	Pneumonectomy	Never	120.00
10	Male	MCE	-	51	VATS	Yes	None	IA	10	Lobectomy	Never	90.00
11	Female	MCE	Low	21	VATS	Yes	None	IA	4	Lobectomy	Never	53.00
12	Male	MCE	Low	15	Open	Yes	None	IA	11	Lobectomy	Never	86.00
13	Male	MCE	Low	54	Open	Yes	None	IB	5	Lobectomy	Never	41.00
14	Female	MCE	-	24	Open	Yes	None	IA	7	Lobectomy	Never	121.00
15	Female	MCE	-	37	Open	Yes	None	IA	4	Lobectomy	Never	90.00
16	Female	ACC	-	52	Open	Yes	None	IB	10	Lobectomy	Never	68.00
17	Male	ACC	High	26	Open	Yes	None	IB	4	Lobectomy	Never	52.00
18	Male	MCE	Low	66	VATS	Yes	None	IA	5	Lobectomy	Present	42.00
19	Female	MCE	-	58	VATS	Yes	Chemotherapy	IB	5	Lobectomy	Never	114.00
20	Female	MCE	-	42	Open	Yes	None	IB	5	Lobectomy	Never	87.00
21	Male	ACC	-	50	VATS	Yes	Chemotherapy	IB	8	Lobectomy	Never	129.00
22	Male	MCE	-	45	VATS	Yes	None	IB	9	Sub-lobectomy	Never	97.00
23	Male	MCE	Low	51	VATS	Yes	None	IA	7	Lobectomy	Never	84.00
24	Female	MCE	Low	20	Open	Yes	None	IB	5	Sleeve lobectomy	Never	49.00
25	Male	MCE	Low	23	VATS	Yes	None	IA	4	Sleeve lobectomy	Never	43.00
26	Male	ACC	-	56	Open	Yes	None	IB	6	Lobectomy	Present	23.00
27	Female	ACC	-	59	Open	Yes	Chemotherapy	IIB	11	Sleeve lobectomy	Never	57.00
28	Male	ACC	-	59	VATS	Yes	None	IA	3	Lobectomy	Never	55.00
29	Male	MCE	Low	70	Open	Yes	None	IA	7	Pneumonectomy	Present	51.00
30	Male	MCE	-	53	Open	Yes	Chemotherapy	IIIA	12	Lobectomy	Present	94.00
31	Female	MCE	Low	29	VATS	Yes	None	IB	3	Lobectomy	Never	41.00
32	Female	ACC	-	54	Open	Yes	None	IB	10	Pneumonectomy	Never	121.00
33	Female	MCE	Low	43	Open	Yes	Chemotherapy	IB	4	Lobectomy	Never	78.00
34	Female	MCE	Low	39	Open	Yes	None	IB	9	Pneumonectomy	Never	46.00
35	Male	ACC	-	39	Open	Yes	Unknown	IB	9	Lobectomy	Never	64.00
36	Female	ACC	-	41	VATS	Yes	Chemotherapy	IB	6	Sleeve lobectomy	Never	79.00
37	Male	MCE	Low	59	VATS	Yes	Chemotherapy	IB	9	Lobectomy	Never	77.00
38	Female	MCE	Low	51	Open	Yes	None	IB	5	Lobectomy	Never	53.00
39	Male	ACC	-	65	VATS	Yes	None	IIA	6	Sleeve lobectomy	Never	22.00
40	Male	MCE	-	27	Open	Yes	Unknown	IIA	7	Lobectomy	Never	87.00
41	Male	ACC	-	65	Open	No	Chemotherapy + radiotherapy	IIIB	7	Sleeve lobectomy	Present	15.00
42	Male	ACC	-	52	VATS	Yes	Chemotherapy + radiotherapy	IIB	7	Lobectomy	Present	61.00

MEC: mucoepidermoid carcinoma, ACC: adenoid cystic carcinoma, DFS: disease-free survival, VATS: video-assisted thoracic surgery. Cases 26, 35, and 42 had lung cancer with bilateral lung metastases. Case 39 had lung recurrence and liver metastasis. Case 41 had bone metastasis.

**Table 3 cancers-14-04668-t003:** Univariable and multivariable Cox proportional hazard regression analyses for prognostic factors in lung cancer patients with ADC, SCC, and SGC.

	Univariable Analysis		Multivariable Analysis		
Variables	HR	*p*-Value	HR	95% CI	*p*-Value
Sex					
Male	1		1	reference	
Female	0.771	<0.001	0.798	0.791–0.805	<0.001
Grade					
Well	1		1	reference	
Moderate	1.487	<0.001	1.298	1.271–1.326	<0.001
Poor	2.160	<0.001	1.471	1.440–1.503	<0.001
Undifferentiated	3.233	<0.001	1.354	1.326–1.383	<0.001
Tumor location					
Upper lobe	1		1	reference	
Middle lobe	0.988	0.276	1.020	0.998–1.042	0.076
Lower lobe	1.023	<0.001	1.045	1.035–1.056	<0.001
Other location	1.860	<0.001	1.181	1.153–1.208	<0.001
Unknown	2.061	<0.001	1.167	1.151–1.183	<0.001
Age (year)					
≤64	1		1	reference	
>64	1.214	<0.001	1.220	1.209–1.232	<0.001
Histological types					
ADC	1		1	reference	
SCC	1.161	<0.001	1.119	1.108–1.130	<0.001
SGC	0.361	<0.001	0.513	0.437–0.603	<0.001
Chemotherapy					
No	1		1	reference	
Yes	1.086	<0.001	0.516	0.511–0.521	<0.001
Radiotherapy					
No	1		1	reference	
Yes	1.373	<0.001	0.905	0.897–0.914	<0.001
Unknown	1.285	<0.001	0.907	0.863–0.953	<0.001
Marital status					
Non-married	1		1	reference	
Married	0.848	<0.001	0.880	0.872–0.888	<0.001
Unknown	0.901	<0.001	0.893	0.874–0.913	<0.001
Race/ethnicity					
Caucasians	1		1	reference	
Others	1.053	<0.001	0.898	0.889–0.908	<0.001
Unknown	0.518	<0.001	0.574	0.481–0.685	<0.001
Surgical approaches					
None	1		1	reference	
Sub-lobectomy	0.300	<0.001	0.499	0.489–0.510	<0.001
Lobectomy	0.215	<0.001	0.347	0.342–0.353	<0.001
Pneumonectomy	0.345	<0.001	0.424	0.408–0.441	<0.001
Unknown surgical approaches	0.447	<0.001	0.520	0.464–0.582	<0.001
Unknown	1.117	0.002	0.907	0.845–0.973	0.007
TNM stage					
IA	1		1	reference	
IB	1.193	<0.001	1.320	1.292–1.349	<0.001
IIA	1.571	<0.001	1.620	1.567–1.674	<0.001
IIB	1.588	<0.001	1.975	1.931–2.020	<0.001
IIIA	2.192	<0.001	2.501	2.453–2.549	<0.001
IIIB	3.193	<0.001	2.978	2.917–3.041	<0.001
IV	5.734	<0.001	4.787	4.707–4.869	<0.001

OS: overall survival, HR: hazard ratio, CI: confidence interval, ADC: adenocarcinoma, SCC: squamous cell carcinoma, SGC: salivary gland-type carcinoma. Cox regression’s method was Enter selection.

**Table 4 cancers-14-04668-t004:** Univariable and multivariable Cox proportional hazard regression analyses of postoperative prognostic factors in salivary gland-type carcinoma patients.

	Univariable Analysis		Multivariable Analysis		
Variables	HR	*p*-Value	HR	95% CI	*p*-Value
Sex					
Male	1		1	reference	
Female	1.137	0.616	1.117	0.649–1.924	0.689
Grade					
Well to moderate	1		1	reference	
Poor to undifferentiated	5.317	<0.001	3.809	2.103–6.897	<0.001
Tumor location					
Upper lobe	1				
Middle lobe	0.707	0.462			
Lower lobe	1.017	0.957			
Other location	0.798	0.595			
Unknown	1.069	0.888			
Age (year)					
≤64	1		1	reference	
>64	3.924	<0.001	4.043	2.202–7.423	<0.001
Chemotherapy					
No	1		1	reference	
Yes	2.583	0.009	1.306	0.533–3.200	0.560
Radiotherapy					
No	1		1	reference	
Yes	2.273	0.009	1.283	0.579–2.842	0.540
Unknown	10.85	0.020	1.553	0.146–16.51	0.715
Marital status					
Non-married	1				
Married	0.806	0.411			
Unknown	1.470	0.601			
Race/ethnicity					
Caucasians	1				
Others	0.645	0.205			
Surgical approaches					
Sub-lobectomy	1		1	reference	
Lobectomy	0.564	0.068	0.439	0.225–0.856	0.016
Pneumonectomy	1.334	0.424	0.487	0.195–1.216	0.123
N classification					
N0	1		1	reference	
N1	2.939	0.001	3.486	1.535–7.917	0.003
N2	4.139	<0.001	2.103	0.744–5.945	0.161
Tumor size (cm)					
1.0–3.0	1		1	reference	
3.1–5.0	1.905	0.027	2.409	1.180–4.915	0.016
>5.0	3.244	0.001	1.677	0.662–4.244	0.275
Histological subtypes					
MECs	1		1	reference	
ACCs + EMC	1.931	0.010	0.845	0.450–1.584	0.599

HR: hazard ratio, CI: confidence interval, MEC: mucoepidermoid carcinoma, ACC: adenoid cystic carcinoma, EMC: epithelial-myoepithelial carcinoma. The race/ethnicity of two patients was unknown. The Cox regression method was Enter selection.

## Data Availability

Any researchers interested in this study should contact the correspondence author Dong Xie to request the data.

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
