# Peer review of "The Clinical and Prognostic Characteristics of Primary Salivary Gland-Type Carcinoma in the Lung: A Population-Based Study"

_cancers, 2022, doi:10.3390/cancers14194668_

Round 1
Reviewer 1 Report (Previous Reviewer 1)
The manuscript has improved based on reviewers' comments and can be accepted in this form
Author Response
Dear reviewer,
Thanks for your timely and professional review.
Reviewer 2 Report (Previous Reviewer 2)
I have read and reviewed the revised manuscript and answers to my queries. The now entitled manuscript “The clinical and prognostic characteristics of primary salivary gland-type carcinoma in the lung: a population-based study” is much improved. Nearly all the points have been answered. A few additional points of clarification may be of benefit.
Revision Points:
1. Line 85 of the marked version, which corresponds to Minor point #2 in the original review, the authors state “the TNM stage was re-translated.” This does not appear to be clarified, as the authors state in the response to reviewer. What does “re-translating” the TNM stage entail?
2. Line 86 the pharse “due to a lack” is duplicated. Please edit.
3. Back to the topic of “retranslation”, do the authors mean that cases were restaged from earlier versions of the AJCC staging manual to the current one? Again, very confusing.
4. Line 110, consider “the majority of patients” in lieu of what was written (“the major part of the patients”).
5. Consider putting the 95% CI into the text of the manuscript all the time and not just some of the time.
6. Interesting how both chemotherapy and radiotherapy were harmful in Univariate analysis and helpful in Multivariate analysis. Any thoughts?
7. Line 181, “was not reached” would be better than “did not reachedhed” (and reached was misspelled).
8. Line 217, there is an extra “s”. Please edit.
9. Line 257, LNR is used without defining this term in the manuscript.
Author Response
Dear reviewer,
Thanks for your constructive suggestions and professional review. Your useful comments and suggestions helped us improve the quality of the manuscript. Thank you, sincerely. Besides, we have modified the manuscript based on your suggestions.
Revision Points:
- We are sorry about it. And, we have clarified this sentence again.
- We have edited it.
- We have revised it again.
- According to your suggestion, we have modified it.
- We have added 95% CI into the text of the manuscript.
- You indicated an interesting phenomenon. In our points, radiotherapy and chemotherapy are related to more advanced-stage diseases. For example, patients with stage I-IIA majorly receive operations as routine treatment; however, patients with IIIB majorly receive chemotherapy and radiotherapy. Therefore, chemotherapy and radiotherapy are harmful to patients in the univariable analysis. After adjusting for other confounders, such as the TNM stage, chemotherapy and radiotherapy are helpful to patients in the multivariable analysis. Thank you.
- We have modified it.
- We have revised it.
- We have modified it. Thank you.
This manuscript is a resubmission of an earlier submission. The following is a list of the peer review reports and author responses from that submission.
Round 1
Reviewer 1 Report
The data reported in this article represent a large-scale retroactive study and the difficulty that the reader encounters is to find an effective correlation between all the parameters analyzed. In the future it will be necessary to carry out further studies to confirm the hypotheses reported in this study.
However, it is appreciable and there are some small changes to be made.
Minor comments
· All abbreviations must also be made explicit in the abstract
· Figure 3 has a low magnification to be displayed; it is necessary to reinsert it at a higher magnification.
Reviewer 2 Report
I have read and reviewed the manuscript “The incidence and surgical significance to the prognosis for primary salivary gland-type carcinoma in all lung cancers: a population-based study” from Dr. Wu and colleagues. This is a large SEER database analysis of a rare tumor with a concurrent look at their own institutional experience, although it is unclear over how many years. While the concept is good, there are a number of limitations and concerns that are raised that I hope the authors will be able to address in a revision of the manuscript.
Major Points:
1. In the ABSTRACT the authors should make it know they utilized the SEER database.
2. The graphical abstract might be improved by cutting out the table under the Crude incidence rate to make the graph larger (along with then adding a legend). The other side, consider just one of the 2 K-M curves. Also consider a single summary point. I highly recommend: https://www.elsevier.com/authors/tools-and-resources/visual-abstract
3. In the INTRODUCTION the authors aim to “investigate the morbidity and surgical significance of the prognosis”. I can find no evidence of an evaluation of morbidity in this manuscript. Do the authors mean “incidence”? If so, these two terms are not quite interchangeable and may confuse the reader as many surgeons refer to morbidity as complications (excluding death/mortality) from surgery.
4. Statistically, instead of excluding patients “lost follow-up within 60 months”, should these be censored? This needs to be answered by the statistician. Also, can the authors please provide in their response to reviewers (it does not need to go in the manuscript) who was the statistician from the 8 authors and what is their statistical expertise?
5. In the RESULTS, the “cohort for incidence analysis” appears to be all 593,662 patients. Is this correct? This gives an incidence of 0.08%, compared to 0.11% when only using the 219,011 “eligible patients”. Is this difference significant? How can the authors account for this difference? This then brings the remainder of the descriptive statistics into question. Also, just because a patient dies within a month does not mean they should be excluded from analysis of incidence.
6. One of the points that keeps bothering me is from the SEER database how do we definitively know that these patients were not metastatic SGC? If this is unknown then the issue of incidence is muddied.
7. Why exclude patients who died within a month of diagnosis? Is this to exclude stage IV patients? But, then stage IV patients are taken into account as seen in Table 1.
8. Calculations of OS is biased due to the exclusion criteria. These cases should have been censored, not excluded.
9. The fact that there appears to be a statistically significant difference between stage IA ADC and SCC is concerning. Also, the 5-year OS for stage IA ADC is about 70%, which is within published literature. However, for SCC it is under 50%. This raises concern and should be discussed.
10. I don’t see the overall survival of the 42 patients from Shanghai in Table 2. How was OS survival calculated? I cannot seem to find how many patients from Shanghai met criteria of having primary pulmonary SGC and did not undergo surgery. Can the authors direct me to this in the manuscript? Also, what is the column “Combined stage”? It appears to just be the stage accounting for the TNM criteria.
11. If the authors wish to discuss the issue with “confounders” in the paper by Zhu et al, they should inform the reader of which confounders they did not consider.
12. Line 285 states that there were 52 ACC, 113 MCE, and 1 EMC in the “present study”. But, there were only 42 in the surgery cohort of the Shanghai data and 248 in the “eligible” patients from SEER. From where is this new number of 166 patients? Also, the graph with “subpopulation cutpoints” only has 162 patients. These numbers need to all be check and either made uniform or explained why they are different in the manuscript.
Minor Points:
1. In the ABSTRACT, “surgical significance” is unclear. As the paper looks into the role of surgery on outcomes, it might be more appropriate to consider saying such as, “…incidence rate and role of surgery to the prognosis…”.
2. I’m sorry, but I don’t understand the following statement: “According to the 8th TNM staging system [13], the combined stage of eligible patients for survival analyses was re-translated.” Can the authors please clarify?
3. The next sentence is also unclear. Why were cases from 2016-2018 needed to be “re-staged”? Because they were excluded do to a “lack of re-staging details.”
4. In Figure 1, exclusion criteria 4 could be better worded. Consider listing actual histologies that were excluded other than a double negative of excluding all histology types [except] excluding adeno, squamous, and salivary-type. Also, should salivary-gland-type all be hyphenated?
5. Can the authors explain how “Marital status” impacts OS?
6. The table of the 42 patients could easily be a supplemental table. Also, surgical approach is more likely to be “Open” or “VATS”. Consider changing the title of the column currently “Surgical Approach” to “Type of Resection”.
7. “Logistic” does not need to be capitalized in “Logistic regression” in section 3.2.
8. Table 2 Legend – can the authors clarify “lung-metastasis diseases”.
9. The authors should note in the limitations that a “sub-lobar” resection is poorly defined and includes both wedge and segmentectomy.
10. “in the lobe”, I believe the authors mean “lobectomy”. (line 280)